# Sensitivity of Planaria to Weak, Patterned Electric Current and the Subsequent Correlative Interactions with Fluctuations in the Intensity of the Magnetic Field of Earth

**Victoria Hossack \*, Michael Persinger and Blake Dotta \***

Behavioral Neuroscience, Laurentian University, Sudbury, ON P3E 2C6, Canada; mpersinger@laurentian.ca
\* Correspondence: vx_hossack@laurentian.ca (V.H.); bx_dotta@laurentian.ca (B.D.)

**Abstract:** Some species of fish show highly evolved mechanisms by which they can detect exogenous electric and magnetic fields. The detection of electromagnetic fields has been hypothesized to exist in humans, despite the lack of specialized sensors. In this experiment, planaria were tested in a t-maze with weak electric current pulsed in one arm to determine if the planaria showed any indication of being able to detect it. It was found that a small proportion of the population seemed to be attracted to this current. Additionally, if the experiment was preceded by a geomagnetic storm, the planaria showed a linear correlation increase in the variability of their movement in response to the presence of the weak electric field. Both of these results indicate that a subpopulation of planaria show some ability to respond to electromagnetic fields.

**Keywords:** flatworm; geomagnetic storms; Ap index; electroreception

---

## 1. Introduction

There are several species of fish that have specialized cells for detecting electric fields in their environment [1–3]. Most fish capable of eletro- or magnetoreception do so by producing a weak electromagnetic field of their own and detecting fluctuations and disturbances of this endogenously produced field [2]. Some electroreception receptors involved operate through low-voltage L-type calcium (Cav1.3) and potassium ion channels [3]. The Cav1.3 channels are present in a wide variety of other cells, not traditionally thought to have the capacity of electroreception, including dopamine secreting cells in the brain [4,5] and cochlear cells in the ear canal responsible for hearing [6]. While land mammals do not live in a medium as electrically conductive as water, air is able to efficiently conduct magnetic fields [7]. Additionally, electromagnetic fields are generated by organs whose cells communicate with each other with bioelectric discharges caused by the movement of ions across the cell membrane, such as in the heart [8] and in the brain [9]. Some experiments have indicated the potential for reception of disturbances of the Earth's magnetic or electric field. For example, the application of a magnetic field with the same intensity as the Earth's was used to classically condition sharks [1]. The potential interaction between environmental electromagnetic fields has already been demonstrated in the human brain, where the pattern of its electrical activity showed transient periods of coherence with the Earth's magnetic field [10,11].

Planaria are small, fresh water flatworms that have central nervous systems containing all of the classical neurotransmitters that are found in humans [12]. In the 1960's, Frank Brown Junior demonstrated that planaria could reliably re-orient themselves relative to the North–South compass direction, indicating they may be able to detect the Earth's static magnetic field [13,14]. The present

experiment aimed to investigate electroreception in planarian flatworms. We did this by placing the planaria in a maze that had an electric field in one arm of the maze, and then observing the planaria's movement for indications of a preference towards—or avoidance of—any specific arm.

## 2. Methods

### 2.1. Planaria

Planaria (*Dugesia tigrina*) were housed in a fridge at 4 degrees Celsius. It should be noted that housing planaria at such a low temperature is not the norm in planaria care [15]. However, planaria have been found living in the wild at these temperatures [15,16], and we have found this method to be effective at preventing mass mortalities. They were fed calf liver weekly. Before any experimentation began, they were given 10–15 min to acclimatize to room temperature and were not used within 3 days of being fed. They were kept at all times in President's Choice spring water. All experiments were completed between 10 am and 10 pm.

### 2.2. Electric Field Application

The electric field was generated by a homemade system consisting of a SainSmart ATMega2560 microcontroller (SainSmart Technology, Inc., Lenexa, KS, USA) that interfaced with a circuit built onto a breadboard that consisted of a diode, transistor, and 20 kΩ resistor. The breadboard also had an R2R resistor ladder, which allowed digital to analog conversion of the electric field. The digital input originated from a Gateway laptop (model: NV53A) computer. From this computer, Arduino software programming was used to construct the parameters of the field.

The field used is called Thomas, patterned after a chirp in a communication system [17] (Figure 1). This pattern was created through the Arduino-microcontroller output system, which translated points to a potential difference between +/− 5 V. The duration of each point was set to 3 ms (milliseconds), and the delay between the end of one cycle through the pattern and the beginning of another was also set to 3 ms. The amount of time for one cycle through the pattern was ~ 2.5 s. The electric field was turned on before the planaria were placed in the maze for the field group, and was left off for the entire duration of the sham group.

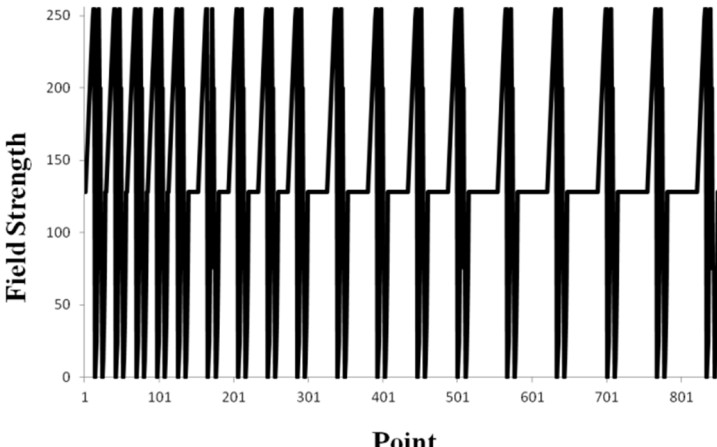

**Figure 1.** Thomas pattern, modeled after chirp features of a communication system.

A picture of the t-maze used can be seen in Figure 2. It was made from a rectangular plastic dish filled with paraffin wax, which had a "T"-shaped mould. A schematic of the t-maze is found in Figure 3. A pair of electrodes (one positive and one negative) were placed opposite one another on Line B, in either Arm 1 or Arm 2. The electrodes were separated by about 1 cm and were able to conduct electricity to each other through the spring water.

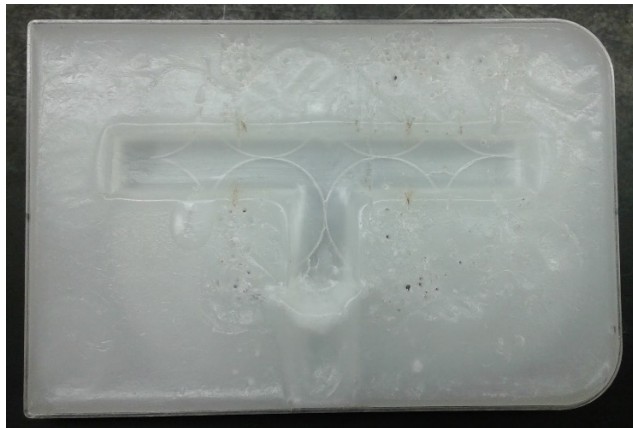

**Figure 2.** A picture of the t-maze that was used for this experiment. It was created by filling a plastic dish with paraffin wax and molding the "T" shape into this. During testing it was filled with ~7 mL of President's Choice spring water.

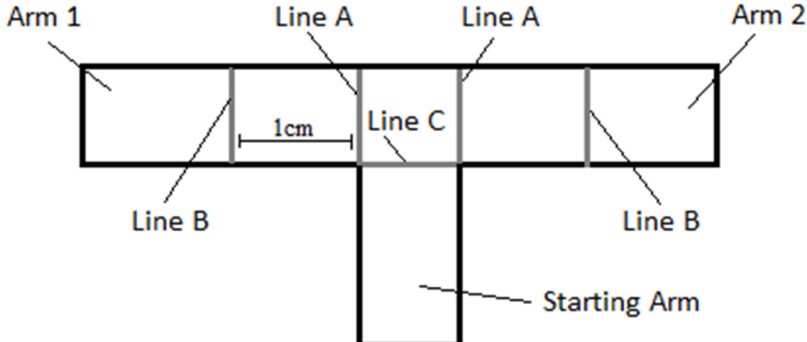

**Figure 3.** Drawing of the t-maze and the layout of all the arms. The electrodes were placed at Line B, in either Arm 1 or Arm 2. The planaria were placed in the bottom of the starting arm and allowed to roam free for 5 min. The arms that they visited and the time for them to cross Line C.

### 2.3. Behavioral Measures

The planaria were placed, one at a time, in the bottom of the starting arm and allowed to roam freely in the maze for 5 min. The movement of the planaria was recorded for the following measurements: the amount of time it took them to leave the starting arm (cross Line C in Figure 3), which arms in the maze they went into, the amount of time it took for them to cross into each arm (cross either of the Line A's in Figure 3), and the total number of arms they visited. A planaria was considered to have crossed into an arm when its full body had crossed over the line. Each planaria was only used once.

### 2.4. Statistical Analysis

For each day that experiments were conducted, there were 3–5 planaria tested in each of the four conditions. For analysis, the 3–5 planaria in each condition were averaged to obtain a mean score to represent each condition (one each day). Additionally, we also looked at the variability within each condition. This was done by taking the standard deviation of the 3–5 planaria in each condition (on each day).

Analysis of variance (ANOVA) was used when looking for main effects between the presence of the electric field and the sham condition. Pearson's r and Spearman's rho were used to correlate planaria behavior with geomagnetic storm activity. Geomagnetic storm activity was quantified using the Ap (planetary) and AA (antipodal) indices; they are both linear measures of the geomagnetic disturbances and have base units of nanoTesla (nT) [18,19]. The Ap index is derived from the kp (planetary) index and has units of nanoTesla. The values reported for these indices are derived as averages of geomagnetic activity from 13 observatories around the globe [18]. The AA (antipodal) index is derived from two observatories, one in the northern hemisphere in the UK and one in the southern hemisphere in Australia [19].

All statistical analyses were completed with SPSS 20 (Statistical Package for the Social Sciences, version 20).

## 3. Results

There were no significant effects for the presence of the field in any of the variables of planaria movement in the maze ($p > 0.05$; Table 1). Over the course of the experiment, it was noted that occasionally a planaria would go up to one of the electrodes and make contact. This was noted 15 different times out of a total of 143 planaria in the study. The behavior of the planaria was not the same each time this occurred; sometimes the planaria would glide over the electrode and seem unaffected, while at other times the planaria would convulse but stay close to the electrode (even if the current was then turned off to allow the planaria to move away). A new variable was computed for the percentage of planaria that were noted to display this type of behavior for each condition on each day. A two-way ANOVA with independent variables of presence of field and side of electrodes indicated a significant main effect for the presence of field ($F(1,35) = 5.67$, $p = 0.023$; $\Omega^2 = 0.14$; Figure 4).

**Table 1.** Results of analysis of variance (ANOVA) in the planaria behavior variables recorded during their maze exploration. SD refers to the standard deviation values calculated for the 3–5 planaria measured for each day an experiment was conducted.

| Planaria Behaviour Measurements | | Sham Mean (SEM) | Thomas Mean (SEM) | F Statement for Effect of Field |
|---|---|---|---|---|
| Percent that entered arm of electrode | | 36.2 (6.29) | 23.0 (6.53) | $F(1,35) = 2.07$, $p = 0.160$ |
| Time to cross Line C (seconds) | Mean | 102.0 (13.9) | 97.8 (11.7) | $F(1,35) = 0.28$, $p = 0.599$ |
| | SD | 45.3 (7.80) | 47.7 (8.55) | $F(1,35) = 0.00$, $p = 0.995$ |
| Time to cross Line A (seconds) | Mean | 119.2 (9.77) | 106.0 (10.7) | $F(1,35) = 1.00$, $p = 0.325$ |
| | SD | 62.9 (7.28) | 54.4 (8.83) | $F(1,35) = 1.66$, $p = 0.206$ |
| Total number of arms visited | Mean | 0.97 (0.09) | 0.84 (0.11) | $F(1,35) = 0.80$, $p = 0.377$ |
| | SD | 0.50 (0.10) | 0.74 (0.13) | $F(1,35) = 1.82$, $p = 0.187$ |
| Percent that touched an electrode | | 4.31 (2.97) | 18.8 (5.27) | $F(1,35) = 5.67$, $p = 0.023$; $\Omega^2 = 0.14$ |

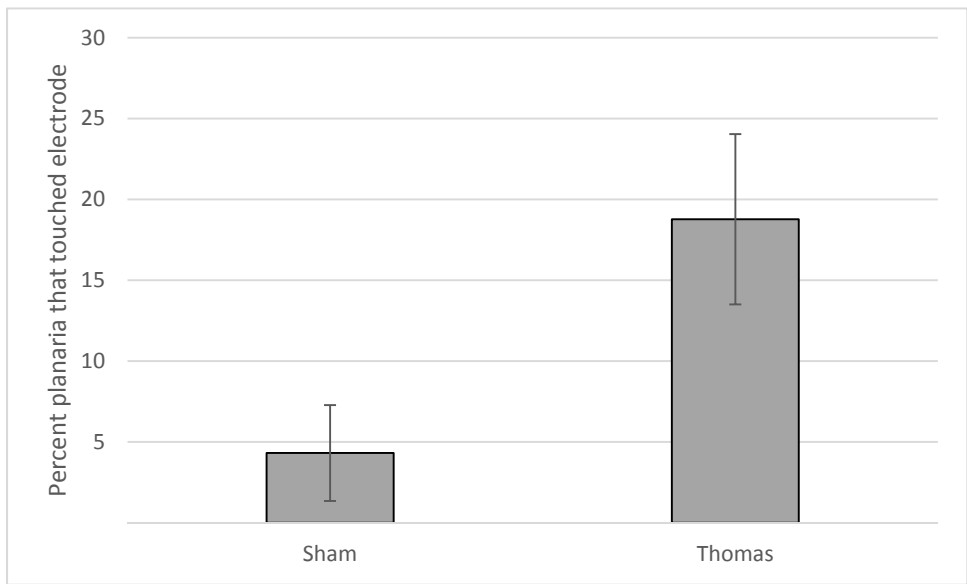

**Figure 4.** Percent of planaria per day of experiment that made contact with an electrode while in the t-maze. There were 19 days of experiments in the Thomas group and 17 days of experiments in the sham group. Error bars represent standard error of the mean (SEM).

Correlational analysis was used to determine any relationships between the continuous variables (the amount of time it took for them to cross Line C or Line A and the number of arms they visited in the 5 min) and the Ap indices. A correlation was considered significant if the Pearson correlation and Spearman correlation analysis were both statistically significant. There were no significant results when the entire dataset was analyzed, but when the groups of Thomas (N = 19 days of experiments) and Sham (N = 17 days of experiments) were analyzed separately, there were significant correlations found with the amount of movement variables. This was quantified as the total number of arms the planaria visited during the 5 min it was given in the t-maze. The mean and standard deviations were taken for the values in each condition for each day of experiment. Results are summarized in Table 2 for the mean number of arms visited and in Table 3 for the standard deviation of the number of arms visited. These tables contain the Pearson and Spearman values; if both these values were found to be significant, then a Fischer's r to z test was used to determine if the Pearson's r was statistically different in the Thomas exposure group compared to the Sham group. The most significant differences were found in the standard deviation values, which are represented in Figure 5. Changes in variability were most likely driving the correlations found with the mean values. No significant correlation values were found with the time (i.e., speed) values ($p > 0.05$).

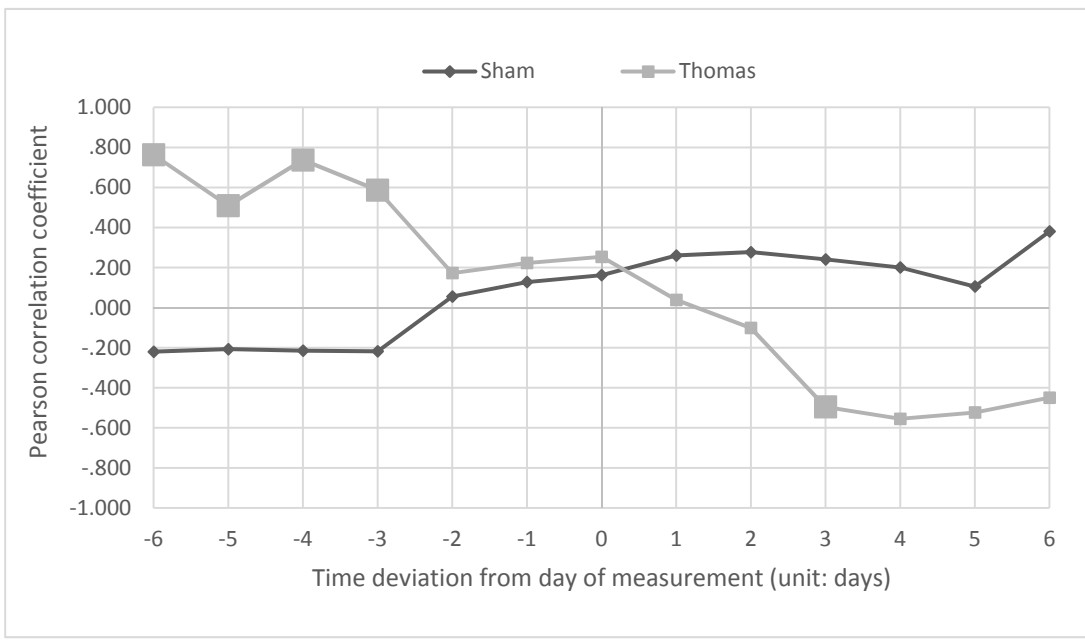

**Figure 5.** Pearson correlation coefficients of the average daily Ap (planetary geomagnetic index) indices on days before and after the day of experiment, with the standard deviation of the total number of arms each planaria visited while in the t-maze. Extra-large data points in the Thomas condition were significant in Pearson's and Spearman's correlation tests and were more than 2 z-scores from their respective point in the Sham group. Ap indices refer to a measurement of fluctuations in the intensity of the geomagnetic field.

**Table 2.** Pearson and Spearman correlation coefficients for the daily average of the total amount of arms visited with the Ap indices of days surrounding the day of experiment. Ap indices refer to a measurement of fluctuations in the intensity of the geomagnetic field.

| Day | Sham | Thomas | Fischer r to z Score |
|---|---|---|---|
| −6 days | r = −0.132; rho = −0.156 | r = 0.476 *; rho = 0.653 * | z = −1.78; $p$ = 0.075 |
| −5 days | r = −0.389; rho = −0.354 | r = 0.283; rho = 0.359 | - |
| −4 days | r = −0.373; rho = −0.304 | r = 0.465 *; rho = 0.449 | - |
| −3 days | r = −0.179; rho = −0.098 | r = 0.478 *; rho = 0.531 * | z = −1.92; $p$ = 0.055 |
| −2 days | r = −0.334; rho = −0.181 | r = 0.151; rho = 0.156 | - |
| −1 day | r = −0.467; rho = −0.272 | r = 0.128; rho = 0.036 | - |
| Day 0 | r = −0.338; rho = −0.058 | r = 0.111; rho = 0.112 | - |
| +1 day | r = 0.170; rho = 0.313 | r = −0.008; rho = 0.042 | - |
| +2 days | r = 0.013; rho = 0.128 | r = −0.218; rho = −0.305 | - |
| +3 days | r = −0.175; rho = −0.137 | r = −0.386; rho = −0.400 | - |
| +4 days | r = −0.024; rho = 0.021 | r = −0.484 *; rho = −0.374 | - |
| +5 days | r = 0.188; rho = 0.196 | r = −0.380; rho = −0.352 | - |
| +6 days | r = 0.060; rho = 0.070 | r = −0.474 *; rho = −0.235 | - |

* = $p < 0.05$.

**Table 3.** Pearson and Spearman correlation coefficients for the daily standard deviation of the total amount of arms visited with the Ap indices of days surrounding the day of experiment. Ap indices refer to a measurement of fluctuations in the intensity of the geomagnetic field.

| Day | Sham | Thomas | Fischer r to z Score |
|---|---|---|---|
| −6 days | r = −0.220; rho = −0.314 | r = 0.763 **; rho = 0.692 * | z = −3.35, *p* < 0.001 |
| −5 days | r = −0.208; rho = −0.201 | r = 0.509 *; rho = 0.461 * | z = −2.11, *p* = 0.035 |
| −4 days | r = −0.215; rho = −0.154 | r = 0.737 **; rho = 0.561 * | z = −3.18, *p* = 0.002 |
| −3 days | r = −0.218; rho = −0.137 | r = 0.587 *; rho = 0.556 * | z = −2.44, *p* = 0.015 |
| −2 days | r = 0.055; rho = 0.085 | r = 0.173; rho = 0.250 | - |
| −1 day | r = 0.128; rho = −0.022 | r = 0.223; rho = 0.225 | - |
| Day 0 | r = 0.162; rho = 0.034 | r = 0.254; rho = 0.324 | - |
| +1 day | r = 0.260; rho = 0.098 | r = 0.039; rho = 0.048 | - |
| +2 days | r = 0.277; rho = 0.192 | r = −0.101; rho = −0.356 | - |
| +3 days | r = 0.241; rho = 0.195 | r = −0.495 *; rho = −0.529 * | z = 2.15, *p* = 0.032 |
| +4 days | r = 0.200; rho = 0.322 | r = −0.555 *; rho = −0.428 | - |
| +5 days | r = 0.106; rho = 0.270 | r = −0.523 *; rho = −0.486 * | z = 1.88, *p* = 0.060 |
| +6 days | r = 0.380; rho = 0.435 | r = −0.450; rho = −0.288 | - |

* = *p* < 0.05, ** = *p* < 0.001.

The correlation between the standard deviation of the number of arms visited in the Thomas exposure group is evident in the scatterplots (Figure 6). Previous research investigating relationships with the geomagnetic storm indices [20] binned their data into 5 nT increments to investigate threshold effects. While there are not enough data points in this dataset for that type of analysis, it seems worthwhile to visually inspect the scatterplots for any conspicuous thresholds. Comparing Figure 6A,B demonstrates the difference in correlation coefficients shown in Figure 6. Figure 6B indicates that during quiet conditions the daily standard deviation in each condition was in the range of approximately 0–1.40, but started increasing at Ap indices greater than 15 nT. Previous research in this laboratory studying the influence of the geomagnetic field determined threshold values to be 20 nT [20–26]; however, it is important to note that these studies used the AA indices of geomagnetic deviations. When the AA indices were entered into the current dataset, the potential 15 nT threshold that was found with visual inspection of the scatterplot was found at 20 nT (Figure 6C).

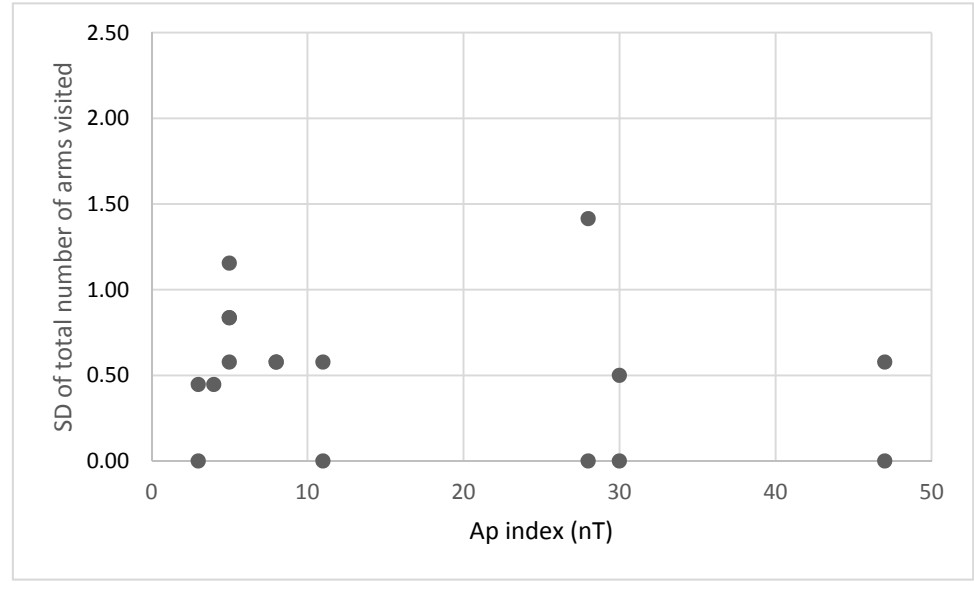

(**A**)

**Figure 6.** *Cont.*

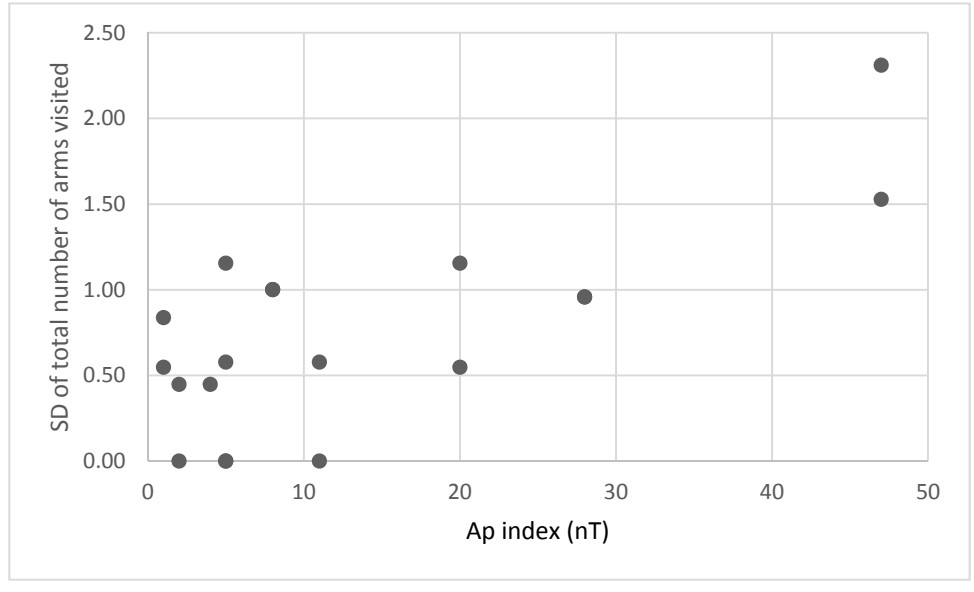

(**B**)

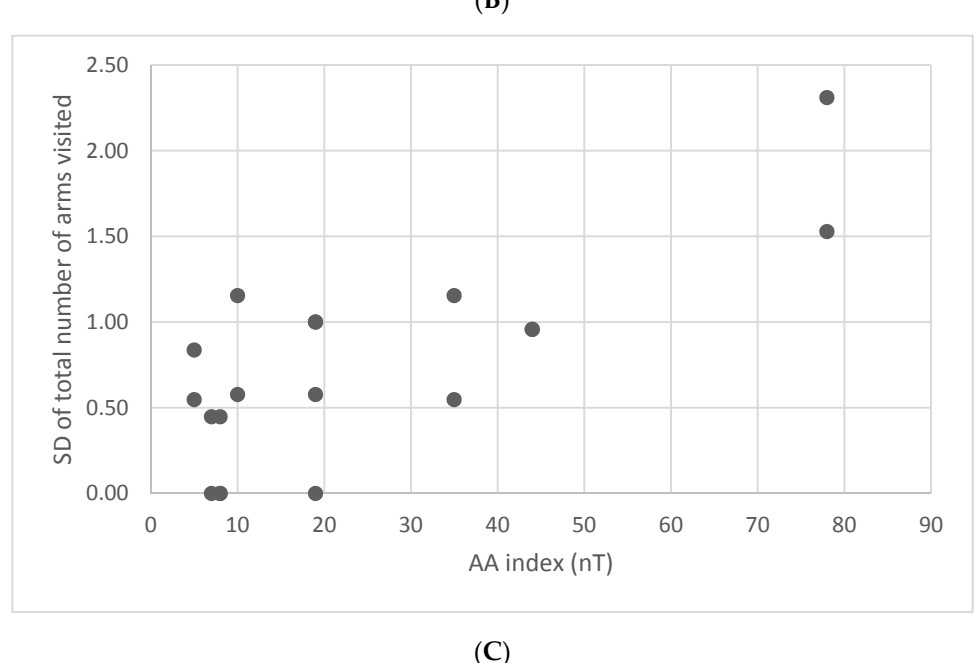

(**C**)

**Figure 6.** Correlation between the standard deviation (SD) in the total number of arms visited by the planaria and the geomagnetic storm indices 4 days before the experiment. (**A**) In the sham-exposed planaria with the Ap index. (**B**) In the Thomas-exposed planaria with the Ap index. (**C**) In the Thomas treated planaria with the AA index. Ap (planetary) and AA (antipodal) indices refer to measurements of fluctuations in the intensity of the geomagnetic field.

## 4. Discussion

This experiment demonstrated that some planaria exhibited positive electro-tatic behavior towards the Thomas-patterned electric field. It should be noted that this electro-tatic behavior was highly variable between planaria observed. For example, some planaria merely glided over an electrode, while others moved towards the electrodes and began convulsing from being in such close proximity but did not show any behavior indicating that they tried to move away. We hypothesize that this variability of responses between planaria is not a genetic feature, but constitutes individual variability of the planaria themselves. Most stayed near the electrode, even when it was turned off. While the

effect was significant, the average proportion of planaria that responded was small; about 18% of the planaria in the Thomas-exposure made contact with an electrode compared to about 4% in the sham condition. This indicates that when the electric field was on, an additional 14% of flatworms made contact with the electrode. Electrotaxis has been demonstrated many times in *C. elegans* [27–29]; however, this may be the first time, or one of the first times, that it has been demonstrated in planaria.

If there were environmental geomagnetic disturbances 3–6 days before the day of the experiment then the planaria exposed to the Thomas-electric field showed an increase in distance travelled as compared to those exposed to sham (Table 2). However, this effect was much more robust when looking at the standard deviation between the distance travelled of the planaria in one group for each day of experiment (Table 3). This indicates that some planaria were more affected than others (if all planaria were affected equally then the strongest effects would have been found in the means, not the variability). Geomagnetic disturbances and electric fields are both forms of electromagnetic fields [7,30]. Exposure to the geomagnetic disturbances may have pretreated the planaria so that they then responded differently to the patterned electric field. Delayed effects of geomagnetic disturbances or storms have been reported before and have been hypothesized to indicate that time was needed to generate the change in signalling pathways that led to the behavior [26]. Additionally, it is interesting to note that the correlations were only found for the number of arms visited (a measure of total movement) but there were no correlations with the time variables (which are a measure of speed). This indicates the geomagnetic storm activity was related to endurance of muscle contractions as opposed to speed of muscle contractions.

Significant trends in various behaviors have been correlated with geomagnetic AA indices in Dr. Persinger's laboratories indicate a threshold of 15–25 nT. These studies include decreased latencies of seizure onset in rats [20], normal rat ambulation and seized rat mortality [22], thyroxine levels in a single patient [25], reports of bereavement hallucinations [21], vestibular experiences [23], out of body experiences [24], and decreased pain thresholds in rats [26].

In the experiment that measured out of body experiences, Dr. Persinger [24] predicted and confirmed that the response of complex human behaviors to increasing geomagnetic storm intensity would be nonlinear. Nonlinear trends were also present in the vestibular experiences [23] and bereavement hallucinations [21] with geomagnetic storm intensities. However, in the experiments with findings of geomagnetic correlations with thyroxine levels [25], pain thresholds in rats [26], and ambulation in normal rats and mortality in seized rats [22], the trends were linear. This may be because these latter behaviors are simpler compared to the complex human behavior described above. These simpler behaviors may have had linear trends if the effects were being driven by primarily one factor, such as a neurotransmitter level.

Future experiments should determine the extent to which the planaria may be electro-tactic. In *C. elegans*, the electrotactic response is so robust that it has been proposed as a method for sorting the organisms as well as determining any that may move abnormally [28]. It has also been determined that the electrotaxis in *C. elegans* is dependent upon functionality of the dopaminergic system [29]. It would be prudent to determine if this is also true for the electro-tactic effect found here with planaria.

**Author Contributions:** Conceptualization, V.H., B.D. and M.P.; methodology, V.H. and B.D.; software, V.H.; validation, V.H., M.P. and B.D.; formal analysis, V.H. and M.P.; investigation, V.H.; resources, M.P. and B.D.; data curation, V.H.; writing—original draft preparation, V.H.; writing—review and editing, V.H. and B.D.; visualization, V.H.; supervision, M.P. and B.D.; project administration, M.P.; funding acquisition, none. All authors have read and agreed to the published version of the manuscript.

**Funding:** This research received no specific grant from any funding agency in the public, commercial, or not-for-profit sectors.

**Acknowledgments:** The authors would like to thank Nirosha Murugan and Trevor Carniello for providing equipment for this experiment.

**Conflicts of Interest:** The authors declare no conflict of interest.

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
