# Peer review of "Sensitivity of Planaria to Weak, Patterned Electric Current and the Subsequent Correlative Interactions with Fluctuations in the Intensity of the Magnetic Field of Earth"

_2571-8800, doi:10.3390/j3010008_

Round 1
Reviewer 1 Report
In this paper Hossack and co-authors investigate whether electric currents can influence the behavior of planarian flatworms. They find correlation between a specific pattern and a specific behavior measurement, as well as relating behavior to geomagnetic disturbances.
I do not believe that the quality of the experiments and their presentation is sufficient to warrant publication. I am highly skeptical of the authors claims as they seem to be engaging in extensive cherry-picking of the investigated variables, do not report all their data and do not make any attempt at describing a mechanism or explanation for the observations. Major changes to underlying experimental approach are needed for this manuscript to contain data of sufficient quality for publication.
Some major issues:
The major reason for concern is that there were a large number of behavioral variables tested and the authors selected only the ones that were significant for presentation. They explicitly talk about selecting certain measurements over others as they showed the desired outcome in the results section. This is blatant p-hacking and unacceptable. Behavioral assays are prone to biases and confounding variables. It is therefore especially important that experimental details are carefully described, and this manuscript is seriously lacking on this front. More experimental details, such as the time of day the experiments were performed, sequence of tests between sham and selected pattern, time between repeat measurement with same animals (across days), light condition, and maintenance conditions of animals between the experiments (incubators etc) were not discussed. Similarly, which of the electrodes in the setup was used for which pattern is unclear. It would be helpful if the authors provided videos of the planarian moving in the T-maze. I am also very surprised that the authors claim to maintain their animals at 4C. I work with animals of the same species and this is not the optimal temperature for them. If this is not a typo, authors need to justify this abnormal temperature. The authors themselves seem unsure of their results and while they describe the observations in their complexity and uncertainty in the beginning of the results section, in the discussion this has morphed into hard statements of facts which are unsupported by the data. E.g.:In Results “…sometimes the planaria would glide over the electrode and seem unaffected, while at other times the planaria would convulse but stay close to the electrode” vs in discussion “This was characterized by the planaria moving towards the electrodes and convulsing from being in such close proximity but not showing any behaviour indicating they tried to move away.”
The authors present a T-maze assay, where the electric field is generated by an electrode in each arm. The authors would need to show that the electric field is not detectable in the other arm, as the size of the whole setup is very small, otherwise the results are very hard to interpret. Furthermore, different patterns at different strengths need to be tested to allow any kind of statement of the link between planarian behavior and electric field presence. Figure 6C is described in the text and the legend but no graph is found in the manuscript. Very little material in the discussion relates to the specific presented data and model system used. The authors should not use the discussion to cite a large number of manuscripts from their own lab, but rather to substantially discuss the mechanisms underlying the described phenomenon.Minor issues:
All data needs to be plotted as dot plots, not as bar graphs
Different axis ranges in 6A and B are confusing and make the comparison between the two conditions needlessly difficult.
In Line 121, authors need to define AP. In the planarian literature commonly used to mean anterior-posterior, but I assume something else is intended here.
Author Response
Thank you for your helpful suggestions. In general, we have heavily cleaned up the methodology section, figure descriptions, and statistical analyses descriptions. With respect to each of your specific concerns:
Minor Issues:
The axes for Figure 6 have been corrected and adjusted to identical ranges. We have also defined Ap in the manuscript.
Major Issues:
We have presented all behavioral assays in a new table, this table reflects all p values - not just the significant ones. We have also increased the clarity of the methodology (this was a key point you addressed, and we feel it's improvement has significantly increased the overall quality of the paper). We have also described the rationale for maintaining the planarias at 4C.
We have also corrected the "morphing in hard statements" issue in the discussion. We feel that the discussion has been significantly improved.
With reference to the "p-hacking" statement, this was not our intention. We only presented the significant results to remain efficient in the manuscript's content. However, given your concerns, we did show all p values for each behavioral assay.
With regards to more electric field testing, this is a phenomenal idea. We believe that the specific experiments and accompanying data from these proposed tests would require a manuscript on their own. The current manuscript has numerous novel findings, and - we feel - is suitable as a first step for this type of research. We agree that additional testing is warranted, but the current data (with the refined discussion & methods section) should be published first.
We thank you for your helpful comments and suggestions. We sincerely feel as though the manuscript is in a much better place after these revisions.
Reviewer 2 Report
The authors investigated the effects of electric current and magnetic fields on planarians. I think that this work is interesting and potentially would be informative for this field. I have three points for the authors’ attention.
1. I do not understand why the subpopulation, not all, of the planarians responds to electromagnetic fields. I wonder regarding the genetic background of the planarians used in this study. Do the authors use clonal planarians?  If the authors do not think that the reason for the variation among individuals is caused by the genetic background, it is better to discuss it more clearly.
2. Methods for many analyses are missing and need to be included both in the methods section and in the figures or figure legends where appropriate. In particular, statistical analyses are puzzled me. What is Ap (AP?), AA, kp, and SPSS?
3. Figure 6C is missing.
Author Response
Thank you for your helpful suggestions. In general, we have heavily cleaned up the methodology section, figure descriptions, and statistical analyses descriptions. With respect to each of your specific concerns:
1] No, these were not clonal planarias. Our hypothesis is that this effect is not due to genetic variation, but most likely individual variability of some unidentified factor. This has been addresses in the text.
2] Clarification on all acronyms has been added
3] Figure 6C has been added.
Round 2
Reviewer 1 Report
The authors have completed revisions in a very timely manner and addressed many of the concerns I raised. I believe that due to the fuller reporting of the results and the more detailed description of the methodology the paper is much improved and can be published as it is now. I thank the authors for their efforts!
I do agree with the authors that future experiments will be necessary to understand the full context of the results presented here. I also understand that these experiments are not feasible within the current publication. I look forward to seeing the authors future work on this topic.